# Serum Dickkopf-1 in Combined with CA 19-9 as a Biomarker of Intrahepatic Cholangiocarcinoma

**DOI:** 10.3390/cancers13081828

**Published:** 2021-04-12

**Authors:** Si-Young Kim, Hee-Seung Lee, Seung-Min Bang, Dai-Hoon Han, Ho-Kyoung Hwang, Gi-Hong Choi, Moon-Jae Chung, Seung-Up Kim

**Affiliations:** 1Department of Internal Medicine, Yonsei University College of Medicine, Seoul 120-752, Korea; kimsy0616@naver.com (S.-Y.K.); LHS6865@yuhs.ac (H.-S.L.); bang7028@yuhs.ac (S.-M.B.); 2Department of Hepatobiliary and Pancreatic Surgery, Yonsei University College of Medicine, Seoul 120-752, Korea; dhhan@yuhs.ac (D.-H.H.); drhhk@yuhs.ac (H.-K.H.); CHOIGH@yuhs.ac (G.-H.C.); 3Yonsei Liver Center, Severance Hospital, Seoul 120-752, Korea

**Keywords:** dickkopf-related protein 1, carbohydrate antigen 19-9, intrahepatic cholangiocarcinoma, diagnosis

## Abstract

**Simple Summary:**

Cholangiocarcinoma (CCC) is a rare cancer, but its incidence and mortality have been increased in the past few decades worldwide, representing a global health problem. CCC is usually asymptomatic in early stages and, therefore, often diagnosed when the disease is already in advanced stages, which highly compromises therapeutic options, resulting in a dismal prognosis. The current diagnosis of CCC by non-invasive approaches such as serum biomarker, carbohydrate antigen 19-9 (CA 19-9), is not accurate enough due to the limitations in its low sensitivity, especially at the early stages of the disease. Therefore, new biomarkers with higher sensitivity and specificity are needed. As the clinical significance of dickkopf-related protein-1 (DKK-1) has been reported in various tumors including intrahepatic CCC (ICC), we aimed to identify the diagnostic and prognostic performance of the DKK-1 and its additive effect combined with CA 19-9 in patients with CCC.

**Abstract:**

Dickkopf-related protein 1 (DKK-1) has a diagnostic and prognostic value in various malignant tumors. We investigated the diagnostic and prognostic performance of DKK-1 in combination with carbohydrate antigen 19-9 (CA 19-9) in cholangiocarcinoma (CCC) patients. Serum DKK-1 levels were measured using enzyme-linked immunosorbent assay. The receiver operating characteristic (ROC) curve, area under ROC (AUROC) analyses, Kaplan–Meier method, and Cox proportional hazard model were used to evaluate the diagnostic and prognostic performance of DKK-1 in combination with CA 19-9. We checked DKK-1 levels in 356 CCC patients and found that DKK-1 was significantly elevated only in 79 intrahepatic CCC (ICC) patients compared to controls (340.5 vs. 249.8 pg/mL, *p* = 0.002). The optimal cutoff level of DKK-1 used to identify ICC patients was 258.0 pg/mL (AUROC = 0.637, sensitivity = 59.5%, specificity = 56.9%, positive predictive value (PPV) = 40.5%, negative predictive value (NPV) = 74.0%, positive likelihood ratio (LR) = 1.38, and negative LR = 0.71). Using this cutoff, 47 (59.5%) patients were correctly diagnosed with ICC. DKK-1 in combination with CA 19-9 showed a better diagnostic performance (AUROC = 0.793, sensitivity = 74.7%, specificity = 56.3%, PPV = 45.7, NPV = 81.8, positive LR = 1.71, and negative LR = 0.45) than CA 19-9 alone. The low DKK-1 and CA 19-9 expression group had a significantly longer overall survival (OS) than the high expression group (*p* = 0.006). The higher level of DKK-1 and CA 19-9 was independently associated with shorter OS (hazard ratio = 3.077, 95% confidence interval 1.389–6.819, *p* = 0.006). The diagnostic and prognostic performance of DKK-1 in combination with CA 19-9 might be better than those of CA 19-9 alone in ICC patients.

## 1. Introduction

Cholangiocarcinoma (CCC) is an epithelial cell malignancy arising from different locations in the biliary tree. It can be classified into intrahepatic, perihilar, and distal cholangiocarcinomas [1]. The prognosis of CCC patients is poor because most patients are diagnosed at an advanced stage [2].

There is no blood biomarker with high accuracy for CCC [3]. The serum levels of known biomarkers such as carcinoembryonic antigen and carbohydrate antigen 19-9 (CA 19-9) are measured to diagnose CCC, but these biomarkers have limitations due to their low sensitivity, especially at the early stages of the disease [3]. Clinicians use CA 19-9 for prognosis prediction, but the false-positive rate is high because of cholangitis [4]. Therefore, new biomarkers with higher sensitivity and specificity are needed.

Dickkopf-related protein 1 (DKK-1) is a member of the Dickkopf family that was identified as a secreted protein and is an inhibitor of Wnt/β-catenin signaling pathway [5,6]. The clinical and prognostic significance of DKK-1 has been reported in breast cancer, lung cancer, esophageal cancer, myeloma, pancreatic cancer, and hepatocellular carcinoma [7,8,9,10,11,12,13,14]. Shi et al. reported that DKK-1 was highly expressed in intrahepatic CCC (ICC) tumor tissues after surgery, and the serum DKK-1 level was significantly higher than that of the control group, indicating a potential role of DKK-1 as a diagnostic surrogate for ICC patients [15].

We investigated the diagnostic and prognostic performance of DKK-1 and its additive effect when combined with CA 19-9 in ICC patients.

## 2. Materials and Methods

### 2.1. Blood Samples and Clinical Information

Blood samples and data used in this study were obtained from the Severance Hospital Biobank and the Korea Biobank Network (The Biobank of Pusan National University Hospital, Pusan, Korea, Gyeongsang National University Hospital, Jinju, Korea, Keimyung University Dongsan Hospital, Daegu, Korea, Chonbuk National University Hospital, Jeonju, Korea, Chungnam National University Hospital, Daejeon, Korea, and Korea Institute of Radiological and Medical Sciences Radiation Biobank, Seoul, Korea, which is affiliated with Korea Cancer Center Hospital, Seoul, Korea, are members of Korea Biobank Network, Cheongju, Korea).

Ninety-two blood samples from CCC patients who underwent surgery were collected from Severance Hospital, Yonsei University, Republic of Korea, between April 2012 and April 2017. Blood samples were obtained prior to surgical treatment and stored at −80 °C in the Severance Hospital Biobank. An additional 264 blood samples from CCC patients and 200 blood samples from healthy individuals were collected from the Korea Biobank Network of hospitals. Blood samples from CCC patients were collected from patients who underwent surgery at Pusan National University Hospital, Pusan, Korea, Gyeongsang National University Hospital, Jinju, Korea, Keimyung University Dongsan Hospital, Daegu, Korea, and Chonbuk National University Hospital, Jeonju, Korea (*n* = 84, 80, 50, and 50, respectively) between February 2011 and February 2019.

Further, blood samples were obtained from healthy individuals who visited the healthcare center of Chonbuk National University Hospital, Jeonju, Korea, Chungnam National University Hospital, Daejeon, Korea, and Korea Cancer Center Hospital, Seoul, Korea (*n* = 50, 50, and 100, respectively) for a health screening check-up between January 2015 and December 2018. Healthy individuals included male and female subjects aged 20–70 years with no past or current clinical abnormalities such as malignancies. Blood samples from the Korea Biobank Network were collected using BD Vacutainer SS Plus Blood Collection Tubes (BD Biosciences, San Jose, CA, USA) from CCC patients prior to surgery and during a health screening check-up involving healthy individuals. Whole blood samples were centrifuged at 3000 rpm at 4 °C for 20 min, and the supernatant was immediately apportioned into 500 µL aliquots and stored at −80 °C in each biobank.

Clinical data and patient characteristics were obtained from electronic medical records of each hospital. Therefore, some clinical information was only available at Severance Hospital. In the case of tumor stage, tumor-node-metastasis staging based on the staging classification of the American Joint Commission on Cancer (AJCC, 8th edition) was assessed postoperatively. Since all patients in this study received surgical treatment, the surgical method was assessed and divided into major (hepatectomy, extended hepatectomy) and minor (wedge resection, sectionectomy, bisectionectomy) liver resection according to extent of anatomical resection [16].

### 2.2. Patient Selection

We checked DKK-1 levels in 356 CCC patients and 200 healthy individuals. Since the median DKK-1 levels in healthy individuals and CCC patients were 249.8 pg/mL and 267.2 pg/mL, (*p* = 0.077) respectively, and given the known association of DKK-1 with ICC, we selected the ICC (*n* = 79) group to further analysis. The ICC group had a significantly higher DKK-1 level than the ECC group (median: 340.5 pg/mL vs. 255.0 pg/mL, *p* = 0.033) and healthy population (median: 340.5 pg/mL vs. 249.8 pg/mL, *p* = 0.002). There was no significant difference between the ECC group and healthy population (median: 255.0 pg/mL vs. 249.8 pg/mL, *p* = 0.377). We, therefore, studied ICC to determine the diagnostic and predictive role of DKK-1 in CCC (Appendix A).

Among the 79 ICC patients, 20 were patients from the Severance Hospital, and the remaining 59 were patients from other hospitals. As a control group, 160 out of 200 healthy individuals were randomly selected corresponding to two times the number of ICC patients (Figure 1).

### 2.3. Measurement of DKK-1 and CA 19-9

DKK-1 levels were measured using the enzyme-linked immunosorbent assay (ELISA) method (human DKK-1 DuoSet ELISA kit). ELISA was performed according to the manufacturer’s instructions (R&D Systems, Minneapolis, MN, USA). Binding of DKK-1 to its receptor (LRP-6) was measured using a functional ELISA. Briefly, microtiter plates were coated with 3 µg/mL of human LRP-6-Fc chimera (recombinant human LRP6/Fc chimera; R & D Systems) prior to the addition of serum samples, and detection was performed using human anti-DKK-1 antibody (R & D Systems). CA 19-9 was measured with the VITROS 3600 Immunodiagnostic System (Ortho-Clinical Diagnostics Inc., Raritan, NJ, USA) using a chemiluminescence immunoassay.

### 2.4. Statistical Analyses

The Mann–Whitney U and Kruskal–Wallis tests were used for comparisons between groups. Receiver operating characteristic (ROC) analysis was used to determine the diagnostic potential of DKK-1, CA 19-9, and a combination of the two markers. To assess whether the combination of DKK-1 and CA 19-9 was better than each marker, we created a new variable predicted probability by using a binary logistic regression equation. Survival curves were estimated using the Kaplan–Meier method, and the differences among curves were assessed using log-rank tests. Univariable and multivariable analyses were based on the Cox proportional hazards regression model for risk factors influencing survival in patients. A *p*-value < 0.05 was considered statistically significant. We set the cutoff value for CA 19-9 as 37.0 U/mL, which is the standard cutoff value [17]. The diagnostic cutoff value for DKK-1 was calculated using the coordinate values of the ROC curve. To determine the optimal cutoff value, we used the criterion for determining the point where the sensitivity is equal to the specificity. This point on the curve is where the product of sensitivity and specificity is maximum [18,19]. Statistical analyses were performed using the SPSS software package (version 20.0; IBM, Armonk, NY, USA). The statistical significance between the ROC curves and the area under the ROC curves (AUROCs) were calculated using MedCalc version 11.1 (MedCalc Software, Mariakerke, Belgium).

## 3. Results

### 3.1. Patient Characteristics

The mean age of the 79 ICC patients was 67.3 years, and the proportion of male participants was 70.9% (*n* = 56). Diabetes and hypertension were identified in 11 (55%) and 15 (75%) patients, respectively. AJCC stages I, II, III and IV were observed in 18 (39.1%), 9 (19.6%), 12 (26.1%), and 7 (15.2%) patients, respectively. The median levels of aspartate aminotransferase (AST), alanine aminotransferase (ALT), and total bilirubin were 41.0 U/L, 41.5 U/L, and 1.01 mg/dL, respectively. The median levels of DKK-1 and CA 19-9 were 381.1 pg/mL and 38.2 U/mL, respectively (Table 1).

### 3.2. Stratification of Cholangiocarcinoma by Etiologies

We stratified CCC patients by its possible etiologies for further analysis. Of the total 356 CCC patients, 95 patients were able to identify the etiology, and they were classified into sporadic (*n* = 75), hepatitis B virus (HBV) (*n* = 5), hepatitis C virus (HCV) (*n* = 1), liver fluke (*n* = 1), liver cirrhosis (*n* = 6), PSC (*n* = 0), and bile duct stone (*n* = 7), respectively, and their median level of DKK-1 and CA 19-9 are described in Appendix A.

### 3.3. Diagnostic Performance of DKK-1 and CA 19-9

To identify diagnostic performance of DKK-1, CA 19-9, and a combination of the two markers, we checked ROC curves and area under the ROC (AUROC) of them. The AUROCs of DKK-1 and CA 19-9 were 0.637 and 0.769, respectively, and when the two markers were combined, the AUROC was 0.793, which was higher than that of the CA 19-9 alone (Figure 2).

The calculated cutoff value for DKK-1 was 258.0 pg/mL, and that for CA 19-9 was set as 37.0 U/mL, which is widely accepted [17]. The diagnostic indices of DKK-1, CA 19-9, and their combination are described in Appendix A. The sensitivity and negative predictive value (NPV) were higher when DKK-1 and CA 19-9 were combined compared to CA 19-9 alone. (sensitivity = 74.7% vs. 49.4% and NPV = 81.8% vs. 79.8%).

### 3.4. DKK-1 in CA 19-9 False-Negative Patients

To determine the additive effect of DKK-1 on ICC diagnosis, the diagnostic yield of DKK-1 was verified. As a result, 20 out of 40 patients with false-negative CA 19-9 were additionally diagnosed with high DKK-1 levels. The diagnostic yield of DKK-1 was 50% (Figure 3).

### 3.5. Prognostic Performance of DKK-1 and CA 19-9

ICC patients were classified into the high and low expression groups based on a DKK-1 level of 258.0 pg/mL and a CA 19-9 level of 37.0 U/mL, respectively. The median survival of the low and high DKK-1 expression groups were 41.7 and 28.0 months, respectively, and there was no significant difference between the groups (*p* = 0.231 by log-rank test). The median survival of the low and high CA 19-9 expression groups were 53.0 and 27.5 months, respectively, and there was no significant difference between the groups (*p* = 0.117 by log-rank test) (Appendix A).

To further identify the prognostic performance of DKK-1 and CA 19-9 separately, a Cox proportional hazard model analysis was performed. The hazard ratio of DKK-1 and CA 19-9 were 1.408 (95% CI 0.802–2.472, *p* = 0.234) and 1.542 (95% CI 0.893–2.660, *p* = 0.120) in univariate analysis and these results were not statistically significant (Table 2).

### 3.6. Combination of DKK-1 and CA 19-9

Since DKK-1 alone did not yield significant results, we checked whether combining DKK-1 with CA 19-9 could predict the prognosis of ICC patients. ICC patients were classified into high expression and low expression groups based on the cutoff values for DKK-1 and CA 19-9. Among the ICC patients, those with both DKK-1 and CA 19-9 levels lower than the cutoff value were assigned to the low expression group, and those with either the DKK-1 or CA 19-9 level higher than the cutoff value were included in the high expression group. Patients in the low expression group had a longer overall survival than those in the high expression group and showed significant difference. (57.5 vs. 27.5 months, *p* = 0.006, log-rank test) (Figure 4).

In the Cox proportional hazard model analysis, the higher level of DKK-1 and CA 19-9 was independently associated with shorter survival (HR = 3.077, 95% CI 1.389–6.819, *p* = 0.006), together with higher level of total bilirubin (HR = 1.084, 95% CI 1.019–1.154, *p* = 0.011) (Table 2).

## 4. Discussion

To date, no blood biomarkers with sufficient diagnostic and prognostic accuracy have been identified in ICC patients. We observed that a novel serum biomarker, DKK-1, when combined with a conventional serum biomarker, CA 19-9, exhibited acceptable diagnostic and prognostic performance, especially in ICC patients. To the best of our knowledge, this is the largest study to investigate the clinical implications of serum DKK-1 in ICC patients.

Although serum DKK-1 has been reported as a useful biomarker in several cancers, its clinical implications are controversial. For example, serum DKK-1 levels are high in patients with cervical, endometrial cancer, and hepatocellular carcinoma, when compared to controls [20,21,22]. In contrast, some studies have shown that DKK-1 expression is downregulated in colorectal cancer, brain tumor, and papillary thyroid cancer [23,24,25]. In spite of these controversies, it has been reported that high DKK-1 levels are significantly associated with CCC. Shi et al. showed that serum DKK-1 level is positively correlated with lymphatic metastasis and is indicative of poor prognosis during ICC, and noted that the optimal cutoff value of DKK-1 for detecting CCC was 2490 pg/mL (sensitivity = 75.7% and specificity = 100%) [15]. Liu et al. reported that a higher expression of DKK-1 in hilar CCC is significantly correlated with hilar lymph node metastasis, and noted that a loss of DKK-1 expression significantly inhibited cancer cell proliferation and migration [26].

Due to the insufficient diagnostic and prognostic performance of single biomarkers such as CA 19-9, attempts to combine multiple serum biomarkers to enhance the overall diagnostic or prognostic performance have been made in several cancers including CCC [22,27,28]. In CCC, serum matrix metalloproteinase-7, soluble fragment of cytokeratin-19, and miR-1537 combined with CA 19-9 showed higher diagnostic performance than CA 19-9 alone [29,30,31]. Similarly, we showed that the AUROC, sensitivity, and NPV of the combined use of DKK-1 and CA 19-9 was higher than that of CA 19-9 alone. Additionally, we found that DKK-1 can identify 50% ICC patients among those with normal CA 19-9 levels. These findings might support the diagnostic performance of serum DKK-1 in ICC patients.

Although the calculated cutoff level of DKK-1 in our study (258.0 pg/mL) was similar to or higher than those of other cancers, such as cervical (314.13 pg/mL), endometrial (46.95 pg/mL), and gastric (31.92 pg/mL) cancers [21,32], it was much lower than the cutoff proposed for ICC in previous studies (2490 pg/mL) [15]. The exact reason for the difference in DKK-1 cutoff value is unclear; however, it may be because all the patients in this study, including those in the validation cohort whose tumor stage was not provided, underwent surgical treatment, which means that they were in the early stage of the disease. It may also be because of different etiologies of the CCC [33]. Some studies have shown that sporadic cholangiocarcinoma is related to the Wnt/β-catenin signaling pathway associated with DKK-1, whereas primary sclerosing cholangitis associated cholangiocarcinoma is primarily related to inflammation-related cytokine and chemokine pathways [34,35,36]. Therefore, further research is needed to establish an optimal DKK-1 cutoff value.

Interestingly, there was no significant association between DKK-1 and ECC in terms of prognosis or diagnosis in our study. While FGFR2 fusions are commonly found in ICC, KRAS mutations are frequent in ECC and gallbladder cancer and significantly associated with poor prognosis. This mutational difference may be related to the difference in DKK-1 levels between biliary tract cancers [37]. In another aspect, embryologically, intrahepatic cholangiocytes arise from bipotent hepatoblasts, and extrahepatic cholangiocytes share an embryologic origin with the ventral pancreas [38]. Rimland et al. reported that cells from intrahepatic and extrahepatic biliary trees could be diverged in response to the canonical Wnt pathway, which is associated with DKK-1. To explain in more detail, they showed that canonical Wnt signaling caused intrahepatic bile duct cells to cease proliferation whereas extrahepatic bile duct cells could be grown over a prolonged period by an experiment using organoid [39]. In addition, Chen et al. revealed that the canonical Wnt pathway was more activated in ECC than in ICC by confirming the protein level in tissue samples [40]. These recent studies show that the environment of intrahepatic bile duct with relatively less activated canonical Wnt pathway may be related to the increased expression of DKK-1. Therefore, the reason why DKK-1 has different associations with ICC and ECC is probably due to the embryologic and carcinogenetic mutational differences. Further experimental studies will be needed in the future and we would like to propose a prospective study on the diagnostic and prognostic efficacy of DKK-1 in CA 19-9 negative biliary tract cancer patients based on this study for clinical development.

We are also aware of the limitations of our study. First, our study was a retrospective study, which led to a potential selection bias. Specifically, patients with relatively early stage CCC who underwent surgical resection might have been recruited. However, we validated our main results using blood samples from ICC patients and healthy individuals from other hospitals that belong to the Korea Biobank Network to minimize the probability of selection bias. Second, there were restrictions on collecting sufficient clinical information for the healthy subjects due to sample acquisition from the biobanks of various hospitals. Since no other co-variables except serum DKK-1 and CA 19-9 were available in the study population, sufficient adjustment for confounders was not performed. Third, the relatively small number of recruited ICC patients from different institutions might be another pitfall of this study. For this reason, it was difficult to access unified and consistent information in clinical data, especially for tumor stage and comorbid diseases, obtained from various hospitals. However, this was necessary to increase the number of patients, as ICC itself is a rare malignancy [41].

## 5. Conclusions

In conclusion, we showed that serum DKK-1 levels are relatively higher in ICC patients than in ECC patients and healthy individuals. The combined use of DKK-1 and CA 19-9 increased the sensitivity and NPV for discriminating ICC patients from healthy individuals. Additionally, combining two markers could help predict the prognosis of ICC patients. Further large-scale studies are warranted to validate our results.

## Figures and Tables

**Figure 1 cancers-13-01828-f001:**
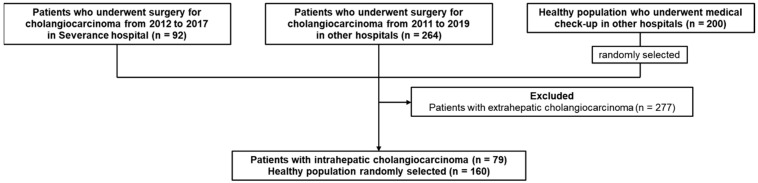
Study flowchart. We collected blood samples of 92 patients with CCC from the Severance hospital. Two hundred and sixty-four blood samples of patients with CCC and 200 blood samples of healthy individuals were collected from other hospitals in the Korea Biobank Network. A total of 277 patients with ECC were excluded and 79 patients with ICC were remained. Of the total 200 healthy individuals, 160 were randomly selected corresponding to two times the number of ICC patients.

**Figure 2 cancers-13-01828-f002:**
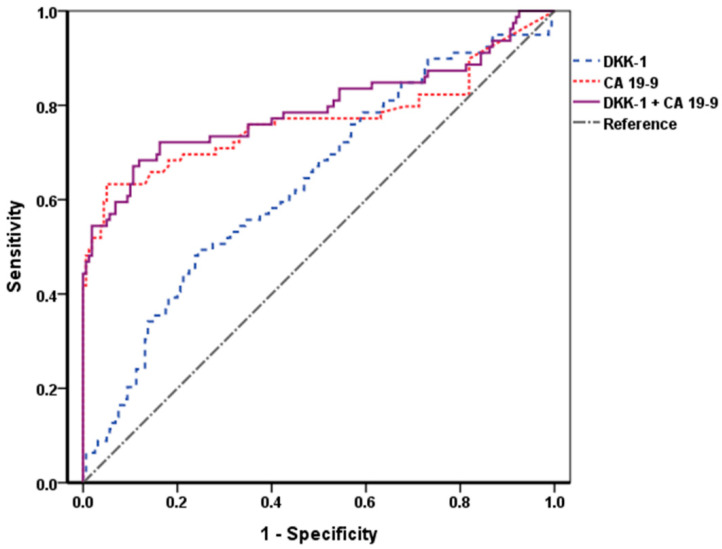
ROC curves of DKK-1, CA 19-9, and their combination. The AUROCs of DKK-1 and CA 19-9 were 0.637 and 0.769, respectively, and the AUROC when the two markers were combined was 0.793. When combined with DKK-1, the AUROC was higher than that of the CA 19-9 alone. DKK-1, Dickkopf-related protein 1; CA 19-9, carbohydrate antigen 19-9; ROC, receiver operating characteristic; AUROC, area under the ROC.

**Figure 3 cancers-13-01828-f003:**
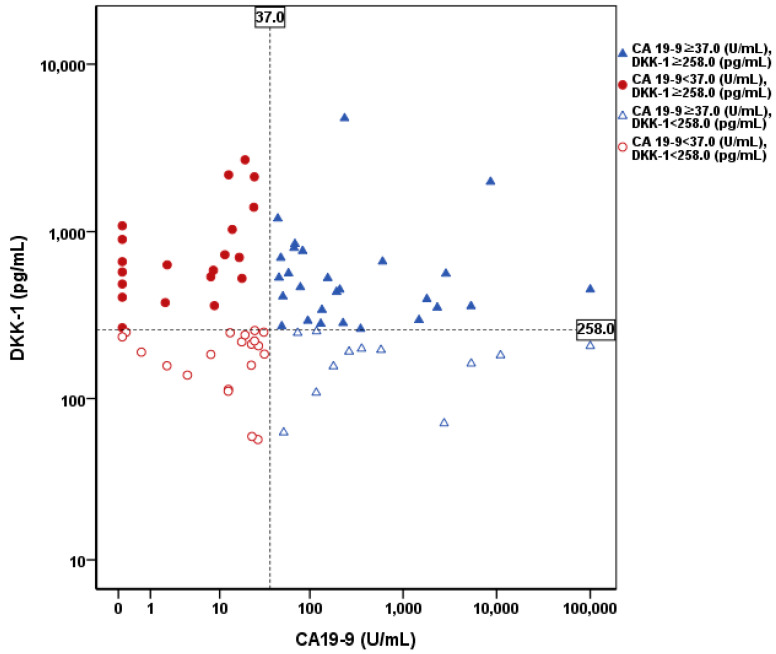
Distribution of DKK-1 and CA 19-9 levels in patients with ICC. The cutoff value of DKK-1 and CA 19-9 were 258.0 pg/mL and 37.0 U/mL, respectively. Twenty out of 40 patients with false negative CA 19-9 were additionally diagnosed using DKK-1. The diagnostic yield of DKK-1 was 50%. DKK-1, Dickkopf-related protein 1; CA 19-9, carbohydrate antigen 19-9; ICC, intrahepatic cholangiocarcinoma.

**Figure 4 cancers-13-01828-f004:**
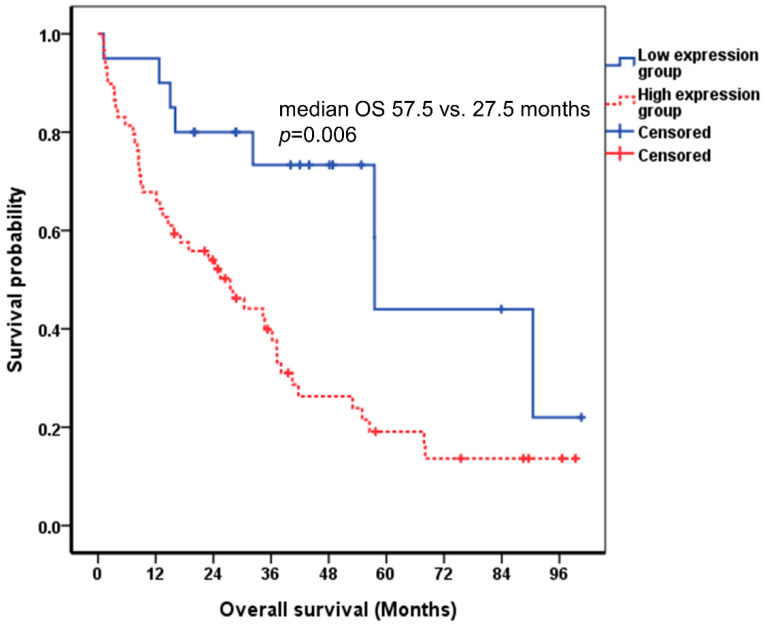
Correlation between DKK-1 combined with CA 19-9 and the survival of patients with ICC using the Kaplan-Meier curve. The cutoff value of DKK-1 and CA 19-9 were 258.0 pg/mL and 37.0 U/mL, respectively. The median survival of the low and high expression groups were 57.5 and 27.5 months, respectively (*p* = 0.006 by log-rank test). DKK-1, Dickkopf-related protein 1; CA 19-9, carbohydrate antigen 19-9; ICC, intrahepatic cholangiocarcinoma; OS, overall survival.

**Table 1 cancers-13-01828-t001:** Baseline characteristics.

Variables	Patients (*n* = 79)
Male gender	56 (70.9)
Age (years)	67.3 ± 11.0
Diabetes mellitus *	11 (55.0)
Hypertension *	15 (75.0)
Smoking *	
Never/ current or ex	11 (55.0)/9 (45.0)
Alcohol *	
Never/ current or ex	12 (60.0)/8 (40.0)
AJCC stage *	
I	18 (39.1)
II	9 (19.6)
III	12 (26.1)
IV	7 (15.2)
AST (U/L)	41.0 (24.8–147.8)
ALT (U/L)	41.5 (21.8–102.0)
Total bilirubin (mg/dL)	1.01 (0.46–3.14)
DKK-1 (pg/mL)	340.5 (206.9–587.2)
CA 19-9 (U/mL)	32.0 (12.6–193.7)

Variables are expressed as mean ± standard deviation, median (interquartile range), or *n* (%). * Patients without clinical information were not included. AJCC, American Joint Commission on Cancer; DKK-1, Dickkopf-related protein 1; CA 19-9, carbohydrate antigen 19-9; AST, aspartate aminotransferase; ALT, alanine aminotransferase.

**Table 2 cancers-13-01828-t002:** Univariate and multivariate analysis to identify independent predictors for survival in patients with ICC.

Variables	Univariate		Multivariate	
	HR (95% CI)	*p* Value	HR (95% CI)	*p* Value
DKK-1 ≥ 258.0 pg/mL (vs. DKK-1 < 258.0 pg/mL)	1.408 (0.802–2.472)	0.234		
CA 19-9 ≥ 37.0 U/mL (vs. CA 19-9 < 37.0 U/mL)	1.542 (0.893–2.660)	0.120		
DKK-1 ≥ 258.0 pg/mL or CA 19-9 ≥ 37.0 U/mL	2.753 (1.294–5.858)	0.009	3.077 (1.389–6.819)	0.006
(vs. DKK-1 < 258.0 pg/mL and CA 19-9 < 37.0 U/mL)				
Age	1.020 (0.992–1.048)	0.160	1.010 (0.981–1.039)	0.511
Female gender	0.922 (0.510–1.664)	0.787	1.287 (0.654–2.535)	0.465
Stage III and IV (vs. I and II) *	1.828 (0.880–3.800)	0.106		
Major surgery (vs. minor surgery) *^,†^	1.476 (0.469–4.648)	0.506		
AST (IU/L)	1.000 (0.998–1.003)	0.794		
ALT (IU/L)	1.000 (0.997–1.003)	0.904		
Total bilirubin (mg/dL)	1.096 (1.032–1.165)	0.003	1.084 (1.019–1.154)	0.011

Variables are expressed as median (interquartile range). * Patients without clinical information were not included. ^†^ Major surgery includes hepatectomy and extended hepatectomy, minor surgery includes sectionectomy, bisectionectomy, and wedge resection. ICC, intrahepatic cholangiocarcinoma; DKK-1, Dickkopf-related protein 1; CA 19-9, carbohydrate antigen 19-9; AST, aspartate aminotransferase; ALT, alanine aminotransferase; HR, hazard ratio; CI, confidence interval.

## Data Availability

Restrictions apply to the availability of these data. Data was obtained from Korea Biobank Network and area available at https://www.koreabiobank.re.kr (23 April 2020) with the permission of Korea Biobank Network.

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
