# Peer review of "Serum Dickkopf-1 in Combined with CA 19-9 as a Biomarker of Intrahepatic Cholangiocarcinoma"

_cancers, 2021, doi:10.3390/cancers13081828_

Round 1

Reviewer 1 Report

The paper is of interest. However, the correlation between Serum Dickkopf-1 and ICC is not clear. I would discuss differences between different biliary tract cancers (intrahepatic, extrahepatic, gallbladder) with different mutation pattern and try to discuss a possibile correlation with Dickkopf-1. A recent review Cancer Management and Research 2019:11 379–388 may be of help. 

Please discuss also on possibile clinical development of your results...would you suggest a possibile clinical trial? Prospective study? 

Author Response

Response to reviewers’ comments

We were very pleased to know that our manuscript (cancers-1143991) has been subject to opportunity of revision for publication in Cancers. We have carefully considered the valuable comments and suggestions provided by the editor and made great efforts to improve the manuscript accordingly. The followings are point-by-point answers to specific questions raised by reviewer. We hope that the revised version of manuscript could meet the priority required for the publication.

Reviewer 1

The paper is of interest. However, the correlation between Serum Dickkopf-1 and ICC is not clear. I would discuss differences between different biliary tract cancers (intrahepatic, extrahepatic, gallbladder) with different mutation pattern and try to discuss a possible correlation with Dickkopf-1. A recent review Cancer Management and Research 2019:11 379-388 may be of help.

Please discuss also on possible clinical development of your results. Would you suggest a possible clinical trial? Prospective study?

Response: We appreciate the reviewer’s keen comment. There are several possible reasons why the intrahepatic cholangiocarcinoma (ICC) is particularly relevant to Dickkopf-related protein 1 (DKK-1). As the reviewer commented, biliary tract cancers show different mutational profiles depending on the location. While FGFR2 fusions are commonly found in ICC, KRAS mutations are frequent in extrahepatic cholangiocarcinoma (ECC) and gallbladder cancer (GBC) and significantly associated with poor prognosis. (1) This mutational difference may be related to the difference in DKK-1 levels between biliary tract cancers. In another aspect, Chen et al. revealed that the canonical Wnt/β catenin pathway was more activated in ECC than in ICC by confirming the protein level in tissue samples. (percentage of samples with partial Wnt pathway activation was 78% vs. 61.3% in ECC and ICC, respectively. P=0.005) (2) In addition, Rimland et al. showed that canonical Wnt signaling caused intrahepatic bile duct cells to cease proliferation whereas extrahepatic bile duct cells could be grown over a prolonged period by an experiment using organoid. (3) These recent studies show that the environment of intrahepatic bile duct with relatively less activated canonical Wnt pathway may be related to the increased expression of DKK-1, the canonical Wnt pathway inhibitor. Further experimental studies will be needed in the future and we would like to propose a prospective study on the diagnostic and prognostic efficacy of DKK-1 in CA 19-9 negative biliary tract cancer patients based on this study for clinical development. We will register this prospective study on ClinicalTrials.gov. However, since the IRB review has not yet been completed within this revision period, we will register on ClinicalTrials.gov as soon as the IRB review is completed in the future.

Reference

  1. Ghidini M, Pizzo C, Botticelli A, Hahne JC, Passalacqua R, Tomasello G, et al. Biliary tract cancer: current challenges and future prospects. Cancer Manag Res. 2019;11:379-88.
  2. Chen W, Liang J, Huang L, Cai J, Lei Y, Lai J, et al. Characterizing the activation of the Wnt signaling pathway in hilar cholangiocarcinoma using a tissue microarray approach. Eur J Histochem. 2016;60(1):2536.
  3. Rimland CA, Tilson SG, Morell CM, Tomaz RA, Lu WY, Adams SE, et al. Regional Differences in Human Biliary Tissues and Corresponding In Vitro-Derived Organoids. Hepatology. 2021;73(1):247-67.

Reviewer 2 Report

In this manuscript, the authors presented that serum DKK-1 in combination with CA19-9 was useful for the diagnosis of ICC patients.  They also explained DKK-1 was significantly elevated only in ICC patients compared to healthy controls and ECC. This DKK-1 might be a good biomarker for diagnosing ICC, but it still has some problems as indicated below. 

Major

  1. The number of this cohort was too small to divide into 2 groups, training cohort and validation cohort. The number of training cohort was only 20 patients, so they should analyze only the entire cohort. It was meaningless to do multivariate analysis for only 20 patients.

  1. The authors tried to identify independent predictors for survival in patients with ICC, but they have done without staging classification and presence/absence of operation. These factors were known predictors and they should analyze them with these known factors (Table 2).

  1. The authors analyzed some variables for identifying predictors (Table 2). The combination of DKK-1 and CA19-9 was examined, but each factor was not analyzed in this result. They should do DKK-1 and CA19-9 separately. They would not get conclusion without this analysis. 

  1. Shi RY et al. have already shown that high DKK-1 was related to poor prognosis. What is the difference between this manuscript and that paper?

Author Response

Response to reviewers’ comments

We were very pleased to know that our manuscript (cancers-1143991) has been subject to opportunity of revision for publication in Cancers. We have carefully considered the valuable comments and suggestions provided by the editor and made great efforts to improve the manuscript accordingly. The followings are point-by-point answers to specific questions raised by reviewer. We hope that the revised version of manuscript could meet the priority required for the publication.

Reviewer 2

In this manuscript, the authors presented that serum DKK-1 in combination with CA19-9 was useful for the diagnosis of ICC patients. They also explained DKK-1 was significantly elevated only in ICC patients compared to healthy controls and ECC. This DKK-1 might be a good biomarker for diagnosing ICC, but it still has some problems as indicated below.

Point 1) The number of this cohort was too small to divide into 2 groups, training cohort and validation cohort. The number of training cohort was only 20 patients, so they should analyze only the entire cohort. It was meaningless to do multivariate analysis for only 20 patients.

Response 1: We agree the reviewer’s comment. In order to overcome the problem of sample size of our study, we analyzed the entire cohort again without dividing the cohort into training and validation cohorts. As a result, the diagnostic indices of combination of DKK-1 and CA 19-9 were as follow: sensitivity=74.7%, specificity=56.3%, PPV=45.7%, NPV=81.8%, positive LR=1.71, negative LR=0.45. To confirm the prognostic performance of DKK-1 and CA 19-9, we obtained the Kaplan-Meier curve. When ICC patients were classified into high expression and low expression groups based on the cutoff values for DKK-1 and CA 19-9, patients in the low expression group had a longer overall survival (OS) than those in the high expression group and showed significant difference. (57.5 vs. 27.5 months, P=0.006, log-rank test) (Figure 4) In the Cox proportional hazard model analysis, the higher level of DKK-1 and CA 19-9 (HR=3.077, 95% CI 1.389-6.819, P=0.006) and total bilirubin (HR=1.084, 95% CI 1.019-1.154, P=0.011) was identified as independent predictor. (Table 2)

To further analysis, we stratified cholangiocarcinoma (CCC) patients by etiology and their median level of DKK-1 and CA 19-9 were as follows: sporadic (n=75, 218.9mg/dl, 51.3U/mL), HBV (n=5, 219.0mg/dl, 171.3U/mL), HCV (n=1, 70.7mg/dl, 2746.0U/mL), liver fluke (n=1, 318.4mg/dl, 95.4U/mL), liver cirrhosis (n=6, 256.5mg/dl, 203.7U/mL), bile duct stone (n=7, 525.6mg/dl, 89.4U/mL). (Supplementary Table 2) Furthermore, we identified the diagnostic and prognostic performance of DKK-1 for each of these etiologies, but there were no meaningful results, and there was no etiology showing statistically significant difference in median OS according to DKK-1 cutoff value. We added this information in Results section.

In addition, we tried to obtain the data of previous study in order to set it as a validation cohort. To do this, we asked the authors (Shi et al., Cancer, 2013;119(5):993-1003.) of the study several times if they could share the raw data of ICC patients, but we could not receive it, unfortunately. Instead, we will register another prospective study with large sample size on ClinicalTrials.gov. However, since the IRB review has not yet been completed within this revision period, we will register on ClinicalTrials.gov as soon as the IRB review is completed in the future.

Point 2) The authors tried to identify independent predictors for survival in patients with ICC, but they have done without staging classification and presence/absence of operation. These factors were known predictors and they should analyze them with these known factors (Table 2).

Response 2: We appreciate the reviewer’s keen comment. As the reviewer commented, we analyzed ICC patients by adding a stage as a variable in the Cox proportional hazard model analysis, and since all patients in this study received surgical treatment, the surgical method was added as a variable, not the presence/absence of operation. The surgical method was divided into major (hepatectomy, extended hepatectomy) and minor liver resection (wedge resection, sectionectomy, bisectionectomy) according to the anatomical resection range. (1) The result was as follows: stage III and IV (vs. stage I and II) (HR=1.828, 95% CI 0.880-3.800, P=0.106), Major surgery (vs. minor surgery) (HR=1.476, 95% CI 0.469-4.648, P=0.506). These values were the results of univariate analysis and were not significant. (Table 2) We added this information in Methods and Results section.

Point 3) The authors analyzed some variables for identifying predictors (Table 2). The combination of DKK-1 and CA19-9 was examined, but each factor was not analyzed in this result. They should do DKK-1 and CA19-9 separately. They would not get conclusion without this analysis.

Response 3: We appreciate the reviewer’s keen comment. As the reviewer commented, we performed a Cox proportional hazard model analysis to identify the prognostic performance of DKK-1 and CA 19-9 separately. The hazard ratio of DKK-1 and CA 19-9 were 1.408 (95% CI 0.802-2.472, P=0.234) and 1.542 (95% CI 0.893-2.660, P=0.120) in univariate analysis and these results were not statistically significant. (Table 2) We added this information in Results section.

Point 4) Shi RY et al. have already shown that high DKK-1 was related to poor prognosis. What is the difference between this manuscript and that paper?

Response 4: We appreciate the reviewer’s keen comment. The difference from previous study is that this study attempted to identify the diagnostic and prognostic implication of DKK-1 through a noninvasive test. In particular, this study has a distinctive point in focusing on the additional diagnostic ability of DKK-1 in CA 19-9 negative ICC patients. Because, in clinical aspect, CA 19-9 shows a false negative of up to 10%, especially in Lewis negative, so CA 19-9 has limitations in diagnosis and prediction of prognosis in these patients. (2) In the result of our study, 20 out of 40 patients with false negative CA 19-9 were additionally diagnosed using DKK-1 and the diagnostic yield of DKK-1 was 50%. (Figure 3) Through this study, it is shown that DKK-1 could help to compensate the limitations of CA 19-9, the most well-established diagnostic and prognostic biomarker of cholangiocarcinoma. However, due to the limitation in terms of retrospective design of this study, prospective studies are required, so we will register another prospective study on ClinicalTrials.gov. and will recruit additional patients for validation.

Reference

  1. Karamarkovic A, Doklestic K. Pre-resectional inflow vascular control: extrafascial dissection of Glissonean pedicle in liver resections. Hepatobiliary Surg Nutr. 2014;3(5):227-37.
  2. Ballehaninna UK, Chamberlain RS. The clinical utility of serum CA 19-9 in the diagnosis, prognosis and management of pancreatic adenocarcinoma: An evidence based appraisal. J Gastrointest Oncol. 2012;3(2):105-19.

Reviewer 3 Report

In this work, the authors show that the combination of DKK-1 and CA 19-9 can improve to a certain extent the diagnostic prediction of ICC in terms of a limited increase in the AUROC and a more prominent increase in sensitivity than CA 19-9 alone. In addition, they describe a higher detection rate of false negatives. With respect prognosis, they establish that not DKK-1 alone, but the combination of the two biomarkers can predict overall survival. While this is a novel and well-conducted study on very relevant patient samples, in my opinion the study has the following weakness that should be addressed:

- Given the different possible etiologies of CCC, it would be interesting to show the stratification of the patients according to them.

- Since only the ICC subgroup of the cohort is relevant for the study, and no further analyses have been performed for the ECC or the total CCC cohort, I wonder if it is relevant to include the ECC samples in the study. In my view, the knowledge about the lack of implication of DKK-1 as a biomarker for ECC is already established. Maybe further analyses not centered on DKK-1 but on other biomarkers and clinicopathological features could add more information and make the whole population become relevant in the study.

- In the introduction, the authors point out that the previous analysis on the predictive value of DKK-1 for ICC uses a small cohort. However, the current work has a cohort of similar size when excluding the ECC cases.  In addition, they should reword the sentence “Dickkopf-related protein 1 (DKK-1) is a secretory protein found in 65 the embryo of Xenopus. (5, 6)”.

- Looking at the methodology, in my opinion, the discovery cohort is not big enough, as it is evidenced in the results regarding prognosis, when only the bigger validation cohort or the complete cohort gives rise to significant results. Maybe the authors could use the cohort of Shi et al. as a validation cohort if they are to some extent comparable. Also, they should be more explicit when it comes to which blood sample they use, since in the main cohort they have two sampling points and in the others only one. I assume that they are using the samples corresponding to comparable moments. In patient selection, maybe the fact that DKK has been reported as a biomarker of response in ICC specifically supports the rationale for focusing on ICC. Therefore, in my opinion, the authors should reword the following sentence accordingly: “As the median DKK-1 levels in healthy individuals and CCC patients were not significantly different (249.8 pg/mL vs. 267.2 pg/mL, 122 P=0.077), CCC patients were divided into the extrahepatic cholangiocarcinoma (ECC) (n=277) and ICC (n=79) groups to compare DKK-1 levels.”

- In the results, apart from the discrete improvement in the diagnostic prediction model, the prognosis results are lacking relevance for the discovery cohort due to the low size. It is maybe more optimal to analyze the whole cohort and not divide it rather than showing non-significant results if we have significant ones with a bigger sample size. I also would like to see the results of the prognostic ability of CA 19-9 alone in this cohort.

Author Response

Response to reviewers’ comments

We were very pleased to know that our manuscript (cancers-1143991) has been subject to opportunity of revision for publication in Cancers. We have carefully considered the valuable comments and suggestions provided by the editor and made great efforts to improve the manuscript accordingly. The followings are point-by-point answers to specific questions raised by reviewer. We hope that the revised version of manuscript could meet the priority required for the publication.

Reviewer 3

In this work, the authors show that the combination of DKK-1 and CA 19-9 can improve to a certain extent the diagnostic prediction of ICC in terms of a limited increase in the AUROC and a more prominent increase in sensitivity than CA 19-9 alone. In addition, they describe a higher detection rate of false negatives. With respect prognosis, they establish that not DKK-1 alone, but the combination of the two biomarkers can predict overall survival. While this is a novel and well-conducted study on very relevant patient samples, in my opinion the study has the following weakness that should be addressed:

Point 1) Given the different possible etiologies of CCC, it would be interesting to show the stratification of the patients according to them.

Response 1: According to the reviewer’s comment, we stratified cholangiocarcinoma (CCC) patients by etiology and their median level of DKK-1 and CA 19-9 were as follows: sporadic (n=75, 218.9mg/dl, 51.3U/mL), HBV (n=5, 219.0mg/dl, 171.3U/mL), HCV (n=1, 70.7mg/dl, 2746.0U/mL), liver fluke (n=1, 318.4mg/dl, 95.4U/mL), liver cirrhosis (n=6, 256.5mg/dl, 203.7U/mL), bile duct stone (n=7, 525.6mg/dl, 89.4U/mL). (Supplementary Table 2) Furthermore, we identified the diagnostic and prognostic performance of DKK-1 for each of these etiologies, but there were no meaningful results, and there was no etiology showing statistically significant difference in median OS according to DKK-1 cutoff value. We added this information in Results section.

Point 2) Since only the ICC subgroup of the cohort is relevant for the study, and no further analyses have been performed for the ECC or the total CCC cohort, I wonder if it is relevant to include the ECC samples in the study. In my view, the knowledge about the lack of implication of DKK-1 as a biomarker for ECC is already established. Maybe further analyses not centered on DKK-1 but on other biomarkers and clinicopathological features could add more information and make the whole population become relevant in the study.

Response 2: We definitely agree to the reviewer’s comment. According to the reviewer’s opinion, DKK-1 did not show meaningful results in ECC, so we additionally stratified the ECC by etiologies as mentioned above. Nevertheless, there was no etiology that DKK-1 shows significant diagnostic and prognostic implications. Therefore, further research is needed to discover other biomarkers more relevant to ECC, and through this, additional research on the cohort of this study can be proposed.

Point 3) In the introduction, the authors point out that the previous analysis on the predictive value of DKK-1 for ICC uses a small cohort. However, the current work has a cohort of similar size when excluding the ECC cases. In addition, they should reword the sentence “Dickkopf-related protein 1 (DKK-1) is a secretory protein found in the embryo of Xenopus. (5, 6)”.

Response 3: We definitely agree to the reviewer’s comment. In our study, the number of patients is not large enough when ECC patients are excluded. So, we deleted the sentence of Introduction section according to the reviewer’s opinion. We also revised the sentence about defining DKK-1 as follows: Dickkopf-related protein 1 (DKK-1) is a member of the Dickkopf family has been identified as a secreted protein that is an inhibitor of Wnt/β-catenin signaling pathway. (1, 2)

Point 4) Looking at the methodology, in my opinion, the discovery cohort is not big enough, as it is evidenced in the results regarding prognosis, when only the bigger validation cohort or the complete cohort gives rise to significant results. Maybe the authors could use the cohort of Shi et al. as a validation cohort if they are to some extent comparable. Also, they should be more explicit when it comes to which blood sample they use, since in the main cohort they have two sampling points and, in the others, only one. I assume that they are using the samples corresponding to comparable moments. In patient selection, maybe the fact that DKK has been reported as a biomarker of response in ICC specifically supports the rationale for focusing on ICC. Therefore, in my opinion, the authors should reword the following sentence accordingly: “As the median DKK-1 levels in healthy individuals and CCC patients were not significantly different (249.8 pg/mL vs. 267.2 pg/mL, P=0.077), CCC patients were divided into the extrahepatic cholangiocarcinoma (ECC) (n=277) and ICC (n=79) groups to compare DKK-1 levels.”

Response 4: We definitely agree to the reviewer’s comment. In order to overcome the problem of sample size of our study, we analyzed the entire cohort again without dividing the cohort into training and validation cohorts. In addition, as the reviewer commented, we tried to obtain the data of previous study in order to set it as a validation cohort. To do this, we asked the authors (Shi et al., Cancer, 2013;119(5):993-1003.) of the study several times if they could share the raw data of ICC patients, but we could not receive it, unfortunately. Instead, we will register another prospective study with large sample size on ClinicalTrials.gov. However, since the IRB review has not yet been completed within this revision period, we will register on ClinicalTrials.gov as soon as the IRB review is completed in the future.

It seems that the description of the blood sampling timing of ICC patients in the training cohort was ambiguous. The blood samples from ICC patients in the training cohort are comparable to those of the validation cohort because they were obtained once before surgical treatment, not both at the time of diagnosis and surgical treatment. We revised this sentence in Method section to make it clear.

As you mentioned, since it may be considered as if patient selection was made according to the known results for DKK-1 and ICC, the sentence was revised as follows: The median DKK-1 levels in healthy individuals and CCC patients were 249.8 pg/mL and 267.2 pg/mL, (P=0.077) respectively, and we divided the CCC patients into the extrahepatic cholangiocarcinoma (ECC) (n=277) and ICC (n=79) groups to further analysis.

Point 5) In the results, apart from the discrete improvement in the diagnostic prediction model, the prognosis results are lacking relevance for the discovery cohort due to the low size. It is maybe more optimal to analyze the whole cohort and not divide it rather than showing non-significant results if we have significant ones with a bigger sample size. I also would like to see the results of the prognostic ability of CA 19-9 alone in this cohort.

Response 5: We agree the reviewer’s comment. In order to overcome the problem of sample size of our study, we analyzed the entire cohort again without dividing the cohort into training and validation cohorts. When ICC patients were classified into high expression and low expression groups based on the cutoff values for DKK-1 and CA 19-9, patients in the low expression group had a longer overall survival (OS) than those in the high expression group and showed significant difference. (57.5 vs. 27.5 months, P=0.006, log-rank test) (Figure 4) In the Cox proportional hazard model analysis, the higher level of DKK-1 and CA 19-9 (HR=3.077, 95% CI 1.389-6.819, P=0.006) and total bilirubin (HR=1.084, 95% CI 1.019-1.154, P=0.011) was identified as independent predictor. In addition, as the reviewer commented, we performed a Cox proportional hazard model analysis to identify the prognostic performance of DKK-1 and CA 19-9 separately. The hazard ratio of DKK-1 and CA 19-9 were 1.408 (95% CI 0.802-2.472, P=0.234) and 1.542 (95% CI 0.893-2.660, P=0.120) in univariate analysis and these results were not statistically significant. (Table 2) We added this information in Results section.

Reference

  1. Glinka A, Wu W, Delius H, Monaghan AP, Blumenstock C, Niehrs C. Dickkopf-1 is a member of a new family of secreted proteins and functions in head induction. Nature. 1998;391(6665):357-62.
  2. Semenov MV, Tamai K, Brott BK, Kuhl M, Sokol S, He X. Head inducer Dickkopf-1 is a ligand for Wnt coreceptor LRP6. Curr Biol. 2001;11(12):951-61.

Round 2

Reviewer 1 Report

The paper has been significantly improved and is suitable for publication

Author Response

We appreciate the reviewer’s comment.

Reviewer 2 Report

This manuscript has been improved, and became worth reading.

Author Response

We appreciate the reviewer’s comment.

Reviewer 3 Report

Dear authors,

Thank you very much for addressing my comments. In my opinion, this version of the manuscript is evidencing in a clearer manner the impact of combining both DKK-1 and CA 19-9 on the diagnosis and prognosis prediction of ICC.

I still have some remarks regarding several of the points raised in the first revision of the work (highlighted in green):

Point 3) In the introduction, the authors point out that the previous analysis on the predictive value of DKK-1 for ICC uses a small cohort. However, the current work has a cohort of similar size when excluding the ECC cases. In addition, they should reword the sentence “Dickkopf-related protein 1 (DKK-1) is a secretory protein found in the embryo of Xenopus. (5, 6)”.

Response 3: We definitely agree to the reviewer’s comment. In our study, the number of patients is not large enough when ECC patients are excluded. So, we deleted the sentence of Introduction section according to the reviewer’s opinion. We also revised the sentence about defining DKK-1 as follows: Dickkopf-related protein 1 (DKK-1) is a member of the Dickkopf family has been identified as a secreted protein that is an inhibitor of Wnt/β-catenin signaling pathway. (1, 2)

This sentence should be rephrased as following: “Dickkopf-related protein 1 (DKK-1) is a member of the Dickkopf family that was identified as a secreted protein and is an inhibitor of Wnt/β-catenin signaling pathway. (1, 2)

Point 4) Looking at the methodology, in my opinion, the discovery cohort is not big enough, as it is evidenced in the results regarding prognosis, when only the bigger validation cohort or the complete cohort gives rise to significant results. Maybe the authors could use the cohort of Shi et al. as a validation cohort if they are to some extent comparable. Also, they should be more explicit when it comes to which blood sample they use, since in the main cohort they have two sampling points and, in the others, only one. I assume that they are using the samples corresponding to comparable moments. In patient selection, maybe the fact that DKK has been reported as a biomarker of response in ICC specifically supports the rationale for focusing on ICC. Therefore, in my opinion, the authors should reword the following sentence accordingly: “As the median DKK-1 levels in healthy individuals and CCC patients were not significantly different (249.8 pg/mL vs. 267.2 pg/mL, P=0.077), CCC patients were divided into the extrahepatic cholangiocarcinoma (ECC) (n=277) and ICC (n=79) groups to compare DKK-1 levels.”

Response 4: We definitely agree to the reviewer’s comment. In order to overcome the problem of sample size of our study, we analyzed the entire cohort again without dividing the cohort into training and validation cohorts. In addition, as the reviewer commented, we tried to obtain the data of previous study in order to set it as a validation cohort. To do this, we asked the authors (Shi et al., Cancer, 2013;119(5):993-1003.) of the study several times if they could share the raw data of ICC patients, but we could not receive it, unfortunately. Instead, we will register another prospective study with large sample size on ClinicalTrials.gov. However, since the IRB review has not yet been completed within this revision period, we will register on ClinicalTrials.gov as soon as the IRB review is completed in the future.

It seems that the description of the blood sampling timing of ICC patients in the training cohort was ambiguous. The blood samples from ICC patients in the training cohort are comparable to those of the validation cohort because they were obtained once before surgical treatment, not both at the time of diagnosis and surgical treatment. We revised this sentence in Method section to make it clear.

As you mentioned, since it may be considered as if patient selection was made according to the known results for DKK-1 and ICC, the sentence was revised as follows: The median DKK-1 levels in healthy individuals and CCC patients were 249.8 pg/mL and 267.2 pg/mL, (P=0.077) respectively, and we divided the CCC patients into the extrahepatic cholangiocarcinoma (ECC) (n=277) and ICC (n=79) groups to further analysis.

Regarding this last point, my intention is precisely to make it clear that the rationale for selecting the ICC cohort for the study of DKK-1 was indeed the previous knowledge on the association between DKK-1 and ICC. Consequently, I would rephrase that sentence as: “Since the median DKK-1 levels in healthy individuals and CCC patients were 249.8 pg/mL and 267.2 pg/mL, (P=0.077) respectively, and given the known association of DKK-1 with ICC, we selected the ICC (n=79) group to further analysis”

Reference

  1. Glinka A, Wu W, Delius H, Monaghan AP, Blumenstock C, Niehrs C. Dickkopf-1 is a member of a new family of secreted proteins and functions in head induction. Nature. 1998;391(6665):357-62.
  2. Semenov MV, Tamai K, Brott BK, Kuhl M, Sokol S, He X. Head inducer Dickkopf-1 is a ligand for Wnt coreceptor LRP6. Curr Biol. 2001;11(12):951-61.

Author Response

Response: We appreciate the reviewer’s keen comment. As you commented, we revised the sentence defining DKK-1 as follows: Dickkopf-related protein 1 (DKK-1) is a member of the Dickkopf family that was identified as a secreted protein and is an inhibitor of Wnt/β-catenin signaling pathway.

In addition, as your advice, we revised the sentence explaining rationale for selecting the ICC group for study as follows: Since the median DKK-1 levels in healthy individuals and CCC patients were 249.8 pg/mL and 267.2 pg/mL, (P=0.077) respectively, and given the known association of DKK-1 with ICC, we selected the ICC (n=79) group to further analysis